# Characterization of Constitutional Ring Chromosomes over 37 Years of Experience at a Single-Site Institution

**DOI:** 10.3390/genes16070736

**Published:** 2025-06-25

**Authors:** Jaclyn B. Murry, Barbara R. DuPont

**Affiliations:** Greenwood Genetic Center, 101 Gregor Mendel Circle, Greenwood, SC 29646, USA; jmurry@ggc.org

**Keywords:** ring chromosomes, constitutional, mosaic, karyotype, CMA, SNP array, FISH

## Abstract

**Background/Objectives:** Ring chromosomes (RCs) can be rare or common depending on the chromosome involved. With interest in the historical RCs identified by our laboratory, we curated instances to provide further information to this research field. **Methodology**: We carried out a retrospective, single-center study of constitutional RCs identified starting in the late 1980s. Details for 40 RCs with a modal number of 46 chromosomes are featured here. **Results**: Mosaic and non-mosaic RCs are identified, with a preponderance of pediatric-aged females at first ascertainment. We corroborated an enrichment for acrocentric and X chromosome RCs. Six were ascertained perinatally, with peripheral blood being the most commonly studied postnatal specimen type. Notable RCs included a female fetus with an increased risk for monosomy X, whose mosaic RCY arose secondary to a translocation between the sex chromosomes. In another, a series of complex events formed three structurally aberrant chromosomes, including an RC4 with loss of 4p16.3, corresponding with the observed phenotypic expression of Wolf–Hirschhorn syndrome. In another, a mosaic RCX was co-identified with an isochromosome 21q, resulting in a dual diagnosis of trisomy 21 and Turner syndrome. In another, the atypical RC21 structure raises the possibility of a complex rearrangement. Chromosomal microarray data clarified breakpoints and gene dosage imbalances in fifteen, while low-level mosaicism for the RC escaped detection by array in another. Eight RCs were de novo, and parental exclusion was documented for six. **Conclusions**: This study illustrates the need for cytogenomic follow-up to improve the literature for patients with RCs.

## 1. Introduction

Ring chromosomes (RCs) are a type of structural rearrangement that arises from the breakage and fusion of distal regions of the same chromosome. This is a rare abnormality thought to occur in 1 in 50,000 births, and approximately 2800 newborns are born with an RC each year [1]. The first RC to appear in the literature was derived from the X chromosome [2]. Rings have been reported in all chromosomes, with the majority being found in chromosomes 18, 20, and X [3]. Four mechanisms of RC formation have been postulated: (1) fusion of both telomeric or subtelomeric ends of the distal chromosome’s long and short arms, resulting in a complete chromosome ring with little or no loss of chromosome material. (2) Classic full ring formation with double-strand breaks of both distal chromosome arms or one chromosome arm and the telomere or subtelomere of the other chromosome arm, resulting in an RC with loss of material from both the long arm and short arm. (3) Complex chromosome rearrangement involving chromoanagenesis to inverted deletion/duplication rearrangement, resulting in an RC with intrachromosomal duplications and/or deletions. (4) An RC involving nucleolar organizing region (NOR) association for the acrocentric chromosomes, resulting in an RC with a terminal loss of the long arm.

Currently, RCs are diagnosed by karyotype, typically following identification by chromosomal microarray analysis (CMA), based on recognition of loss of both ends of the chromosome. Additional methodologies for defining the stability, size, and completeness of an RC can also include fluorescence in situ hybridization (FISH), whole-genome sequencing (WGS), and optical genome mapping (OGM) [4]. Associated phenotypes range from unaffected to developmental and intellectual delays, along with growth retardation and dysmorphic features. Growth retardation or a “ring syndrome” phenotype has traditionally been linked to dynamic mosaicism and resultant genomic imbalances. Gene dosage imbalances, somatic mosaicism levels, and differences in incomplete penetrance and variable expressivity can further complicate genotype–phenotype correlations. A cytogenetically typical cell line can mitigate unstable RCs and associated phenotypes. Recurrent hotspots are known for certain chromosomes, and gene disruption or positional effects may also modulate the severity of presentation. While familial transmission has been described for some chromosomes, RCs typically arise de novo. Recently, a Chinese human RC registry was published [5], and a new RC database is now available [6], prompting us to curate RCs identified by our cytogenetics team. Our primary aim was to contribute additional confirmed RCs identified by karyotype and, where possible, to clarify genomic imbalances and mosaicism levels, thereby improving our understanding of existing genotype–phenotype correlations for these unique patients. Our secondary aim was to support the future characterization of structural rearrangements, aiming to better understand how these cryptic events mediate RC formation.

## 2. Materials and Methods

### 2.1. Individuals

The inclusion criteria included a retrospective review of de-identified individuals with RCs studied by our laboratory from the late 1980s to the present. Individuals with constitutional RCs were stratified by age at first ascertainment: perinatal (ongoing pregnancy, stillbirths, products of conception), pediatric (0–18 years of age), or adult (>18 years of age). We excluded cancer-associated or supernumerary RCs. Here, we communicate our curation process (Figure 1).

### 2.2. Methods

The laboratory methods included a routine G-banded karyotype, with Q-banded, C-banded, or AgNOR stains performed in a subset of cases. FISH from vendors such as Aquarius, Oncor, and Abbott, Inc./Vysis (Abbott Laboratories, Abbott Park, IL, USA) was conducted on metaphases and/or interphase nuclei. CMA analysis was performed using Custom 22q array, OGT, Cambridge, UK, CytoScan HD, Affymetrix, Santa Clara, CA, USA Genome-wide Human SNP 6.0 Array, Affymetrix, Santa Clara, CA, USA or Global Diversity Array (GDA), Illumina, Inc., San Diego, CA, USA. The genomic coordinates were lifted over to hg19 (GRCh37) to facilitate inter-individual RC comparison. The International System for Human Cytogenomic Nomenclature (ISCN) reflects the version used during analysis. Please see Appendix A for more details.

## 3. Results

### 3.1. Cohort Characteristics

We report constitutional RCs identified at GGC over nearly 40 years. All samples were karyotyped. The number of metaphases examined depended on whether the RC was previously known. Mosaic RC enrichment was observed (Figure 2). Individuals with RCs identified at GGC demonstrated a female preponderance (Figure 2). Upon first ascertainment, six instances occurred perinatally, 24 were pediatric, and 10 were adult-aged. One perinatal instance resulted in a liveborn (Figure 2). In one case, an increased risk for a gonosomal abnormality prompted prenatal cytogenetic studies that incidentally identified an RC; a birth outcome is unknown. Two RCs were identified from products of conception, and another RC was documented in a stillbirth that received an autopsy. The remaining two RC instances identified perinatally lack a known outcome. Typically, blood was analyzed in the postnatal period, and at least five studies were repeated. 

At least fourteen instances underwent FISH, and sixteen had CMA (Figure 3). Other laboratory methods that proved informative in clarifying the RC included subtelomeric MLPA (MRC Holland) testing (n = 2), with X-inactivation studies, and qPCR testing conducted in one each. Eight were de novo, and single parental exclusion occurred in six cases (Figure 3). Tables with reported phenotypes and laboratory findings are displayed at the individual participant level (Table 1 and Table 2). Selected RCs with informative CMA data (based on the UCSC Genome Browser; hg19) are presented in Figure 4, with each panel illustrating different mechanisms for RC formation observed here, along with the partial G-banded karyotype.

### 3.2. Ring Chromosome Summary

Significant RC enrichment was observed for chromosomes X and 22. Additionally, complex rearrangements were detected, resulting in RCs containing centromeric material for chromosome 4 that arose from a three-way translocation involving chromosomes 4, 5, 13, and RC21, which was indicative of chromoanasynthesis, along with an RCY that arose secondarily from a t(X;Y) event. Finally, a mosaic RCX also co-occurred with an isochromosome 21q, resulting in a dual diagnosis for Turner syndrome and trisomy 21 (Down syndrome).

#### 3.2.1. Ring Chromosome 4

We present a case of a newborn female (RC4-1). Testing indications included micrognathia, thin lips, downward slanting palpebral fissures, helix abnormalities, nasal tip anomalies, prematurity, and a congenital heart defect, with advanced bone age, preauricular appendages, an absent or shortened philtrum, microcephaly, midface hypoplasia, and severe intellectual disability appreciated at follow-up. The initial female karyotype obtained from blood found two reciprocal translocations affecting chromosomes 4 and 5 and chromosomes 4 and 13. An RC4 containing 4p12-> 4q12 material was identified, replacing the normal chromosome 4. A follow-up cytogenetic examination using Metaphase FISH employing whole-chromosome paints (Oncor# P5208 chromosome 4 and Oncor# P5208 chromosome 5), and locus-specific probes (D4Z1, which targets the chromosome 4 α-satellite region, and D4S96 associated with Wolf-Hirschhorn syndrome (WHS)), in conjunction with G-banded and C-banded karyotype analyses, clarified the absence of one normal chromosome 4 and the presence of three structurally abnormal chromosomes. The first abnormal chromosome originated from an inverted insertion of a segment of chromosome 4 (4p16.2->4p11) into the long arm of chromosome 5 at 5q35.1. The second abnormal chromosome, identified as a derivative chromosome 13, resulted from the translocation of the long arm of chromosome 4 from 4q21->4qter into the short arm of chromosome 13 at 13p11.2. The third abnormal chromosome was a non-mosaic RC composed of 4p11-> 4q21 material. The results of the FISH analysis utilizing the WHS cosmid probe (Oncor) indicated that D4S96 was absent in the aberrant chromosomes. Meanwhile, the D4Z1 α satellite signal was detected on the abnormal chromosome 5. The complex chromosomal rearrangement (CCR) observed in RC4-1 involved three distinct chromosomes, resulting in a net loss of 4p16.3, consistent with a WHS presentation.

#### 3.2.2. Ring Chromosome 13

We identified four RC13s with a common loss at 13q (RC13-1, RC13-2, RC13-3, and RC13-4), and multiple anomalies were reported, including microcephaly in two individuals. One was de novo, and a maternal origin was excluded in another. Three postnatal cytogenetic cases were tested using blood samples, and one was evaluated in the perinatal setting. C-banding and AgNOR staining were performed in conjunction with standard G-banded karyotyping, which revealed a chromosome 13 origin. Where feasible, at least 30 cells were studied. Both mosaic monosomy 13 and non-mosaic forms were identified. FISH studies were conducted in two using probes hybridizing to chromosome 13; one was informative and underwent CMA (RC13-1). In RC13-2, mild intellectual disability, tonic-clonic (grand mal) seizures, and depression were described, correlating with a non-mosaic karyotype with breakpoints at 13p11.2 and 13q34. RC13-1 and RC13-3 exhibited variation in their breakpoints and karyotypes. The indications for testing in RC13-1 included developmental and speech delay, microcephaly, café-au-lait spots, involuntary gait/movement, sloping forehead, dysmorphic facies, clinodactyly of the fifth finger, and prenatal exposure to toxins or drugs. In RC13-3, only hypotonia and microcephaly were observed. Metaphase FISH studies supported a non-mosaic ring with breakpoints at 13p11.2 and 13q34, corroborated by loss of D13S1825 (Aquarius) in RC13-1. At the same time, CMA (Affymetrix CytoScan HD) analysis for RC13-1 estimated a pathogenic 5.7 megabase (Mb) terminal 13q33.3q34 copy number loss that included 84 genes, among them the *MCF2L*, *UPF3A*, *CHAMP1*, and *SOX1* genes, which have been described as lost in other RC13 cases; maternal exclusion was noted. In RC13-3, a predominant mosaic ring with an unstable dicentric RC13 was found. One stillborn, the product of a second pregnancy (RC13-4), was associated with an abnormal ultrasound and elevated α-fetoprotein (AF-AFP) levels. Initially assessed at 17.4 weeks of gestation, following labor induction, a fetal autopsy revealed holoprosencephaly and multiple congenital anomalies at 23 weeks of gestation. Other features included cystic hygroma, tetralogy of Fallot, hypoplastic external female genitalia, thumb a/hypoplasia, renal a/hypogenesis, choanal atresia, lung a/hypoplasia, and an ectopic anus. The predominantly mosaic RC karyotype, proven de novo, was concordant between amniocytes and placental biopsy and exhibited a co-occurring monosomy for the normal homologue in 6.67~13% of the metaphases examined. A normal FISH study confirmed that the terminal loss at 13q was distal to 13q14 (RB1) (Abbott, Inc./Vysis, Abbott Park, IL, USA) for RC13-4.

#### 3.2.3. Ring Chromosome 14

We identified three females with RC14: one adult affected by moderate intellectual disability and seizures, palpebral fissure abnormality, and a hyperpigmented macule (RC14-1); one product of conception that underwent a postmortem exam (RC14-2); and one child with a seizure disorder (RC14-3). In RC14-2 and RC14-3, a de novo origin was determined. At least 20 metaphases were studied here, and a presumed commonality of 14q loss aiding RC14 formation was identified. Only the ring for RC14-3 was non-mosaic. Breakpoints varied only slightly, ranging from 14q32 to 14q32.3, with mosaicism observed in RC14-1 for loss of the normal homologue (13.8% of metaphases examined). Notably, RC14-2 displayed a single cell with a Robertsonian translocation (14;14) and a normal chromosomal complement, which might represent maternal cell contamination given the RC14’s de novo origin. Alternatively, it could represent true mosaicism for an RC14. CMA (Genome-wide Human SNP 6.0 Array, Affymetrix, Santa Clara, CA, USA) in RC14-1 identified a terminal 464 Kb loss of 14q32.33 in a gene-poor region. In contrast, the CMA (CytoScan HD, Affymetrix, Santa Clara, CA, USA) of RC14-3, affected by seizures, displayed a substantial terminal loss at 14q32.33 (2.57 Mb) of de novo origin. Indeed, *PACS2* was included in this loss, a gene linked to neurological disorders and perturbed in other RC14 cases.

#### 3.2.4. Ring Chromosome 15

We identified a non-mosaic RC15 (RC15-1) in a female product of conception. A total of 51 chromosome spreads were counted. Breakpoints included 15p13 and 15q26 for the de novo RC.

#### 3.2.5. Ring Chromosome 18

Four patients were identified as having an RC18 (RC18-1 through RC18-4). Shared features included developmental delay, failure to thrive, and dysmorphic characteristics. Breakpoint variability was noted with 18p11.2->18p11.31 and 18q22.1->18q23. At least 20 cells were studied in these individuals, with a maximum of 50 metaphases evaluated. One individual’s study included C-banding and routine G-banding to clarify the RC event. One RC18 out of four was non-mosaic (RC18-1). The mosaic forms observed included monosomy for the normal chromosome 18, with or without a normal chromosomal complement. In the latter case, a high proportion (52% of metaphases examined) of a typical cell line might support a post-zygotic origin for the RC and correspond with the observed milder phenotype of cleft palate and FTT, mitigating the severity of his presentation anticipated by RC18 and monosomy 18 (RC18-3). The RC co-occurred with a terminal 18p11.2 deletion in another individual with mosaicism, confirmed by subtelomere 18p MLPA (MRC Holland, Amsterdam, NL) (RC18-2). CMA analysis of RC18-4 (CytoScan HD, Affymetrix, Santa Clara, CA) identified pathogenic terminal losses of 18p11.32->18p11.31 (3.8 Mb) and 18q22.1->18q23 (13.9 Mb); a maternal origin was excluded.

#### 3.2.6. Ring Chromosome 21

Four male children are described here (RC21-1 through RC21-4). Shared features include dysmorphic characteristics, developmental delays, and intellectual disabilities. Three out of four were non-mosaic; the mosaic RC exhibited monosomy 21 for the normal chromosome 21 present in 30% of metaphase spreads. Recurrent breakpoints included 21q22, with half displaying an apparent 21p11.2->21p13 breakpoint. At least two individuals underwent karyotype studies that included G-banding, C-banding, and AgNOR staining, supporting the conclusion that the RC was derived from chromosome 21. RC21-2, RC21-3, and RC21-4 underwent CMA; RC21-4 had a simple deletion of 21q22.13->21qter (9.6 Mb) (OGT oligonucleotide microarray (105k), Cambridge, UK), while another had a simple terminal gain of 21q22.13->21q22.3 (10.2 Mb) (CytoScan HD, Affymetrix, Santa Clara, CA) (RC21-3). The latter underwent subtelomere MLPA (MRC Holland, Amsterdam, NL), which revealed 21q loss and gain of the 21q telomeric sequence, and further internal FISH studies using AML/ETO supported the chromosome 21 origin. One individual (RC21-2) had a complex rearrangement identified by CMA (CytoScan HD, Affymetrix, Santa Clara, CA), showing an overall mosaic loss of the long arm with mosaic interstitial gains of 21q22.11 (4.5 Mb) and 21q22.3 (589 Kb), as well as terminal 21q22.3 loss (4 Mb); maternal inheritance was excluded. Genomic imbalance of the Down syndrome critical region (DSCR) was evident.

#### 3.2.7. Ring Chromosome 22

Fourteen instances of RC22 are detailed here (RC22-1 through RC22-14); seven involve adults, one of whom is deceased, and five are children, one of whom is affected by congenital anomalies and was first identified prenatally through studies involving amniotic fluid (RC22-12), with prenatal screening indicating an increased risk for Down syndrome. An additional fetus (RC22-10) was identified through chorionic villus sampling at 16.1 weeks of gestation; the mother’s prior pregnancy had resulted in a normal male. Recurrent features included intellectual disability and dysmorphic traits. Notably, one individual presented with DiGeorge syndrome (RC22-4), while another was affected by neurofibromatosis (RC22-3). Only two are males. A subset was recognized as part of the GGC’s 22q13 project. Three instances were of de novo origin, with parental exclusion performed in two. Three out of thirteen are mosaic, with the mosaic forms showing a normal chromosomal complement that varied (38% to 92%), suggesting a post-zygotic origin for the RC. A mosaic RC22 co-occurred with a normal chromosomal complement in one case initially identified as a fetus (RC22-12). As a neonate, follow-up karyotype identified a rare cell bearing a 22q13 deletion, with a normal chromosomal complement as the predominant cell population (98% of metaphases examined), raising the possibility of expansion of a cytogenetically normal clone initially identified. In one individual with a non-mosaic RC22, a double pericentric inversion of chromosome 9 was incidentally identified and confirmed by C-banding (RC22-9). Two others also underwent C-banding and/or AgNOR staining as part of their cytogenetic studies, clarifying that the RC was derived from chromosome 22. Five out of six CMA studies (CytoScan HD, Affymetrix, Santa Clara, CA or custom 22q13.31 array, OGT, Cambridge, UK) were informative of *SHANK3* loss, leading to a Phelan-McDermid syndrome-related presentation; in one instance, the CMA was negative due to low-level mosaicism for the ring (8% of metaphases examined). Five individuals experienced a simple terminal loss of 22q13.2->22q31.33 (1.6~7.6 Mb), while one also had interstitial duplication of 22q13.2q13.31 (2.3 Mb) (RC22-13). Five individuals had available informative FISH (Abbott, Inc./Vysis, Oncor, Abbott Park, IL) results supporting terminal long arm copy number changes resulting in RC22.

#### 3.2.8. Ring Chromosome X

Here, we describe seven females with mosaic RCX (RCX-1 through RCX-7). Turner syndrome (TS) was suspected in at least five, with recurrent TS features observed. Six were children. The adult (RCX-2) was diagnosed in early life and had an aberrant right subclavian artery identified incidentally through a CT scan. Her current complaints include hyperlipidemia and premature menopause in her early 20s due to the TS diagnosis. All individuals underwent karyotyping (two with repeat studies), which revealed monosomy X and RCX, with FISH confirming the X-chromosomal origin in three. The available karyotypic breakpoints included Xp22.33->Xp22.2 and Xq25->Xq28. For RCX-3, a total of 100 prometaphase spreads were analyzed, revealing a striking mosaic behavior that we interpret as representing significant instability. Notably, the identified fragment/ring did not fluoresce brightly with Q-banding, but it was positive for C-banding. A follow-up cytogenetic study involving 50 metaphases clarified the incidental dual diagnosis of the X chromosome aberration and isochromosome 21, resulting in trisomy 21 for the long arm and RCX in 4% of metaphases with monosomy X in 92% of the examined metaphases; FISH (Abbott Inc./Vysis, Abbott Park, IL) supported these findings. CMA was attempted for RCX-5, but the resultant data was inconclusive. RCX-6 underwent CMA analysis (CytoScan HD, Affymetrix, Santa Clara, CA), revealing monosomy X and a mosaic status for interstitial X chromosome segments (911 Kb mosaic gain of Xp11.22->Xp11.21 followed by a 25 Mb mosaic loss of Xp21->Xq21.1), consistent with the inability to identify the RC’s origin by karyotype alone; further studies revealed a highly skewed X-inactivation pattern. Genes included in the mosaic gain lacked curation evidence for triplosensitivity. In contrast, the mosaic loss of Xp indicated the loss of *XIST* and conferred an increased risk for various X-linked disorders in this patient. When structurally abnormal X chromosomes lack functional *XIST*, the concern arises for the genome’s inability to inactivate the RC, leading to a more severe phenotype than typically observed in girls affected by TS.

#### 3.2.9. Ring Chromosome Y

A single male child, whose RCY was incidentally found, is reported here (RCY-1). The child presented with a bifid uvula and submucous cleft. Karyotype studies identified a mosaic RCY with karyotypic breakpoints at Yp11.32 and Yq11.21 as a predominant finding (90% of metaphases), with a subset displaying monosomy X (~10%). This finding was supported by CMA (GDA, Illumina, Inc., San Diego, CA), which revealed two pathogenic terminal deletions (1.7 Mb and 302 Kb) affecting: (1) the pseudoautosomal region (PAR1) of the X/Y chromosome, including 16 OMIM genes: *PLCXD1*, *GTPBP6*, *PPP2R3B*, *SHOX*, *CRLF2*, *CSF2RA*, *IL3RA*, *SLC25A6*, *ASMTL*, *P2RY8*, *AKAP17A*, and *ASMT*; and (2) the PAR2 of the X/Y chromosome, which included three OMIM genes: *SPRY3*, *VAMP7*, and *IL9R*. Additionally, an interstitial segment loss of Yq11.21q12, at least 15 Mb in size and of uncertain significance, was identified, encompassing 68 OMIM genes and azoospermia factor (AZF) genes, which are known to be a major cause of male infertility. A paternal origin for this RCY was excluded.

Additionally, amniotic fluid was collected for the cytogenetic study of a female fetus presenting at 16 weeks’ gestational age from a pregnant female whose AFP levels were not elevated (RCX;Y-1). Prenatal cell-free DNA screening indicated a high risk for monosomy X. Concurrent FISH studies on direct amniocytes initially suggested an abnormal pattern indicative of mosaicism for sex chromosome aneuploidy, featuring both monosomy X and an identified XY complement. The mosaic karyotype, assessed from 50 metaphases, revealed a complex event involving a derivative ring Y, likely resulting from an unbalanced translocation between chromosomes X and Y. The second line demonstrated monosomy X present at levels comparable to the RC. FISH analysis for the *STS* locus confirmed the localization of Xp material to the RCY. CMA (GDA array, Illumina, Inc., San Diego, CA) performed on cultures supported a mosaic level for the derivative RCY containing Yp11.32q11.221 (with terminal Ypter and Yqter arm losses of 17 Mb and 11.9 Mb observed, respectively) (~50%), showing a gain of Xp22.33p22.2 (11 Mb) (~50%). These events were interpreted as pathogenic. The patient and pregnancy were ultimately lost to follow-up.

## 4. Discussion

This single-site study involves 40 instances of constitutional RCs. We summarized and compared our cohort findings with those from the existing literature.

There are at least 47 instances of RC4 in the literature [7]. Although our case may not be the most appropriate for comparison with other pure linear imbalances and/or complete RC4 cases, it reflects a unique occurrence resulting in a non-mosaic RC4. Our patient displayed WHS. Publications featuring CMA recount large RCs, with recurrent breakpoints observed at 4p15->4p16 and 4q34->4q35. As nearly complete RC4s may correspond with typical neurodevelopment, it is fitting that our patient exhibited severe disability with losses distal to 4p16 and 4q21->4qter. Mitotic instability is noted in RC4 literature, and poor growth might be attributed to “ring syndrome.” In the absence of a non-mosaic ring, the precursor event may have involved a meiotic and sporadic three-way complex interchromosomal rearrangement, resulting in end-to-end fusion of linear derivative 4 into a ring to promote stabilization. Indeed, patients with larger deletions, including the WHS critical region at 4p, can also present with typical syndromic features. RC4 containing larger 4q deletions can also show variable presentation in a rare subset. Most instances of RC4 are de novo; to our knowledge, familial follow-up was not performed on this individual, but given its complexity, we would presume it was de novo.

Eighty cases of RC13 are described in the literature, showcasing a wide variety of genomic imbalances [8]. Indeed, our patients displayed multiple anomalies, including microcephaly. Dynamic mosaicism exists among carriers, and successive bridge–breakage–fusion cycles can lead to the formation of derivatives and variants of RC13, consistent with at least one observed here. Most cases exhibit incomplete rings, and the corresponding dosage imbalances involve different genes, aligning with our findings. Sixteen published cases have undergone CMA; we include one more with a simple terminal 13q loss containing previously described candidate genes: *MCF2L*, *UPF3A*, *CHAMP1*, and *SOX1*. The loss of the *RB1* gene confers an increased risk for retinoblastoma, but our cases did not show *RB1* involvement. Typically, RC13 arises de novo, although rare familial transmission does occur. Here, our RC13 cases were either de novo or not maternally derived. There is an association between RC13 and an underlying Robertsonian translocation, and family studies should be considered.

RC14 is extremely rare [9]. Simple genomic imbalances may only partially explain the clinical severity. Seizures, developmental and growth delays, facial dysmorphisms, hypotonia, retinal abnormalities, and intellectual disability characterize RC14. Recurrent seizures often develop in infancy and frequently resist medication. Other features include short stature, microcephaly, puffy hands and feet, and recurrent infections [10]. The neurodevelopmental features of RC14 overlap with those of our participants. Indeed, one child here was affected by seizures, whose simple terminal loss included *PACS2*, a candidate gene implicated in other RC14 cases with seizures, which likely arose secondarily to double-strand break repair. RC14 instability is reported, and monosomy for the abnormal RC14 has been observed. The phenotypes associated with RC are more severe than those observed with linear genomic imbalances. Losses at 14q32.2 associated with differentially methylated regions are postulated to arise due to dynamic mosaicism and may confer an additional role for uniparental disomy (UPD) in the RC14 phenotype. It should be noted that a gene-poor deletion in this cytoband was observed in one case, corresponding to a clinically normal array result, negating a simple dosage imbalance. An association between RC14 and an underlying Robertsonian translocation exists, as supported by a single cell observed in the POC. Therefore, family studies should be considered for other potential RC14 cases. Here, we confirmed a de novo origin for two instances of RC14.

More than 100 cases of RC15 are described [11]. The features and genomic events are highly variable; this historical case was not evaluated using more current cytogenomic methods to provide a genomic comparison. Since chromosome 15 is imprinted, the possibility of compensatory rescue for the loss of RC15 should be considered. Most cases are de novo, and our case was confirmed as such. Indeed, the possibility of UPD and its contribution to the RC15 phenotype should be considered in these cases.

Over 110 individuals are described with RC18 [12]. Precise genotype–phenotype correlations for RCs are confounded by factors such as the extent of deletion in RC formation, RC instability, and mosaicism. While rare, it is one of the most common constitutional RCs and may be better tolerated. Patients with RC18 may present features resembling those of the 18p-syndrome, the 18q-syndrome, or a combination of both [13]. The phenotype of the 18q-syndrome includes intellectual disability, hypotonia, microcephaly, short stature, minor facial features, and abnormal male genitalia. At the same time, 18p-syndrome patients exhibit speech delay, short stature, midline defects including holoprosencephaly, a short neck, and IgA deficiency. A reported case of mosaicism for both RC18 and monosomy 18 (11% of cells) displayed short stature, cleft palate, and mild cognitive impairment [14]. Indeed, the prognosis for carriers is improved when holoprosencephaly or heart anomalies are not observed. Wide phenotypic variability arises from differences in genomic imbalances, mitotic instability for RC18, and structural variations in the absence of repetitive sequences mediating rearrangement. This theme is echoed in the participants’ reports herein. Some carriers have a normal line, while others coexist with monosomy 18 due to mitotic instability and a corresponding distal loss of the terminal regions of RC18, or more complex forms of RC18, as described elsewhere in the literature. This aligns with our findings. CMA analysis is recommended to confirm the breakpoints of RC18. Other considerations may include fragile sites or regions that are recurrently susceptible. Although RC18s typically arise as de novo events, maternal transmission of rings from carrier to offspring has been reported; we excluded maternal origin in one.

RC21 has a wide range of features. Dynamic mosaicism is described, along with corresponding genomic imbalances and RCs [15]. To date, 16 carriers have been documented. Two individuals who underwent CMA analysis here exhibited simple terminal copy number changes, and one suggested a complex structural rearrangement resulting in an RC21, all of which were interpreted as pathogenic CNVs. Reports of individuals with mosaicism for monosomy 21 and RC21 are rare; considerable phenotypic variability exists due to differences in breakpoints and genomic content. However, commonly reported phenotypes vary from normal or mildly affected to clinical features, including dysmorphic traits, neurodevelopmental disorders, and congenital abnormalities [16,17]. The presence of multiple gains and losses within the same arm of the chromosome raises the possibility of constitutional chromoanasynthesis, a type of CCR characterized by alternating copy number changes, which include a combination of deletions, duplications, and triplications arising from defective replication, typically clustered on a single chromosome or a few chromosomes [18]. Reports of chromoanasynthesis involving chromosome 21 are scarce; therefore, genotype–phenotype correlation is challenging. However, the clinical presentation may include a combination of features associated with these copy number changes. Both familial and de novo cases of RC21 are documented in the literature. Here, a maternal origin was excluded in two. Carrier females face risks for adverse perinatal outcomes. There are no established hotspots for RC21. Concerns about acute leukemias due to chromosome 21 abnormalities may necessitate follow-up, which is not noted herein.

Over 250 cases of RC22 are described [19], with most displaying PMS clinical features and a subset showing Cat-Eye Syndrome (CES) or DiGeorge syndrome (DGS) presentations. An equivalent gender distribution is often noted, and most arise de novo. A reported preponderance of female carriers might suggest potential sub/infertility for male carriers of RC22. Here, we demonstrate three instances of de novo RC22 and two with paternal exclusion; most carriers in this group were female. Ring instability and dynamic mosaicism are recurrent in RC22. Instability in a balanced and stable RC carrier, typical in their presentation, can result in an affected offspring. RC22 carriers benefit from karyotyping, as CMA may miss the RC configuration. Growth delay occurs more frequently in patients with RCs than in those with linear 22q13 deletions. Variability in deletion size or secondary events on the aberrant 22 is attributed to the PMS phenotypic spectrum. Indeed, we observed copy number losses encompassing the 22q13 deletion syndrome region (PMS). Haploinsufficiency of the *SHANK3* gene is associated with the major neurological features of this syndrome (DECIPHER v11.30). Durand et al. demonstrated that a pathogenic variant in a single copy of *SHANK3* on chromosome 22q13 results in language and/or social communication disorders [20]. Genotype–phenotype analysis conducted in affected individuals highlights the phenotypic variability observed [21]. Indeed, we observed copy number losses encompassing the 22q13 deletion syndrome region (PMS), and our patients displayed PMS-associated features including dysmorphic features, intellectual disability, and developmental delay. While DGS was only seen in one case here, a DGS or CES diagnosis should be considered in atypical cases. Additionally, in one individual with RC22, we identified a double pericentric inversion of chromosome 9, which, as a single event, is a recognized polymorphism and is not considered clinically significant. However, the inversion typically involves only one chromosome 9, and it is somewhat unusual for both homologues to be inverted. In general, cancer predisposition testing is needed to rule out an increased risk for neurofibromatosis type II and atypical teratoid rhabdoid tumors (AT/RT). There is one report of germline RC formation by chromoanagenesis via WGS.

A total of 112 RCX cases are published [22]. Often diagnosed postnatally, RCX presents with features similar to TS, with estimates of approximately 15% of individuals with TS being RCX carriers, and mosaicism with a 45,X cell line is common. The clinical presentation of individuals with this variant TS karyotype varies based on the level of mosaicism, the genomic content missing from the RCX, and whether the RCX is active or inactive, which may lead to functional disomy for genes retained on the RCX. A published comparison between individuals with pure monosomy X and those with RCX indicates that the latter group shows an increased risk for learning difficulties and behavioral maladjustment. Carriers may not show hallmark signs of lymphedema at birth. An enrichment in lower verbal and nonverbal abilities is also noted in RCX compared to 45,X females. Phenotypic groupings include TS-like or a more severe spectrum (learning difficulties, syndactyly, hypotonia, and more significant intellectual disability). Here, most individuals exhibited features similar to those of TS. Indeed, abnormal prenatal screening (increased risk for 45,X) might result in early identification of RCX carriers. CMA and karyotype studies clarify mosaicism levels and address concerns regarding functional X disomy. The loss of *XIST* for structurally abnormal X chromosomes results in the inability to inactivate the RC, leading to a more severe phenotype than typically observed in girls affected by TS. A large and active X chromosome might be associated with lower IQ levels than those of an individual with a small RCX. X-inactivation studies may also be considered; one individual studied here showed significant skewing due to the structurally aberrant RCX. Somatic mosaicism levels can also vary in other body locations. All participants exhibited mosaicism with monosomy X. In rare instances, maternal transmission with spontaneous sexual development occurs; however, parental follow-up was not pursued in our cases.

Sixty-three individuals affected by RCY have been reported [23]. Carriers may present with infertility, short stature, ambiguous genitalia, gonadal dysgenesis, or even TS resulting from the loss of Y chromosome material. Precise genotype–phenotype correlations for RCY are confounded by factors such as the extent of deletion in RC formation and the instability of RCY. In our case, the aberration was discovered incidentally in a male. The Y chromosome contains extensive palindromes, with mirror-image gene pairs necessary for spermatogenesis (AZF regions). These repetitive sequences can act like fragile sites that undergo breakage and rearrangement, resulting in a structurally abnormal Y with significant variability in gains or losses. While typically de novo, rare paternal transmission is also possible; in this case, it was a de novo event. Monosomy X should prompt an investigation for a Y abnormality, especially if a male phenotypic sex is noted. Clarifying the fraction of affected cells is reasonable, as gonadoblastoma risks exist in individuals with monosomy X and the presence of Y marker material in the context of gonadal dysgenesis. The percentage of monosomy X and the genes deleted from the Y chromosome also influences the clinical presentation. A comprehensive genotype–phenotype study beyond the PAR1 region, which contains *SHOX* (Yp11.2) and *AZF* genes (Yq12), is warranted.

Furthermore, mosaic RCY is associated with highly variable clinical presentations, ranging from apparently normal males with azoospermia to mixed gonadal dysgenesis and growth hormone deficiency [24,25]. Most instances of reported RCY occur in a mosaic 45,X/46,X,r(Y) karyotype. RCY containing X-derived material is rarely documented. One report describes a female infant with variant TS, whose ultrasound showed nuchal thickening and postnatal evaluation revealed a low posterior hairline along with moderate edema on the hands and feet. Cytogenetic analysis indicated an approximately 50% mosaic RC composed of material derived from Yp and Yq and from Xp22.1 to Xp21.3 [26]. The intact X chromosome was not subject to random X inactivation. Notably, the percentage of monosomic cell lines in different tissues likely contributes to the broad clinical spectrum reported. Additionally, we found one more female fetus whose nearly equivalent cell populations displayed a mosaic derivative RCY containing Yp (*SRY*) and its centromere, with Xp22.33p22.2 material replacing Yq, as demonstrated by CMA. This line co-occurred with a second line exhibiting monosomy for the second sex chromosome. The parental origin was not evaluated in this study. While not the most appropriate comparator with other simple RCY cases, this perinatal instance is included here to document the atypical and rare findings.

The limitations of this study include the variability of patient-level details due to changes in laboratory information systems and the availability of remnant specimens. Laboratory research interests in abnormalities involving chromosome 22 may underlie our RC22 enrichment. Patients presenting to the clinic and receiving a laboratory work-up may exhibit a more significant presentation. Some patients have been lost to follow-up, precluding conclusions related to natural life history and longitudinal outcomes.

With the introduction of emerging genomic methodologies to elucidate structural rearrangements, individuals with historical RCs would benefit from the refinement of breakpoints and consideration of gene disruption, position effects, and complex dosage imbalances beyond gross karyotypic breakpoints. Future studies of RCs should aim to incorporate and reconcile these newer methods in conjunction with previously available cytogenetic data.

## 5. Conclusions

Karyotyping for RCs remains essential as it reveals mosaicism, the RC structure, and secondary instability at the single-cell level. Cytogenomics aids in detecting genomic imbalances in studies involving pooled genomic DNA. Phenotypic variability exists even within the same RC, making it challenging to predict the clinical picture. In summary, continued study of the underlying mechanisms, the role of resulting genomic changes, and cell dynamics will enhance our understanding of RCs.

## Figures and Tables

**Figure 1 genes-16-00736-f001:**
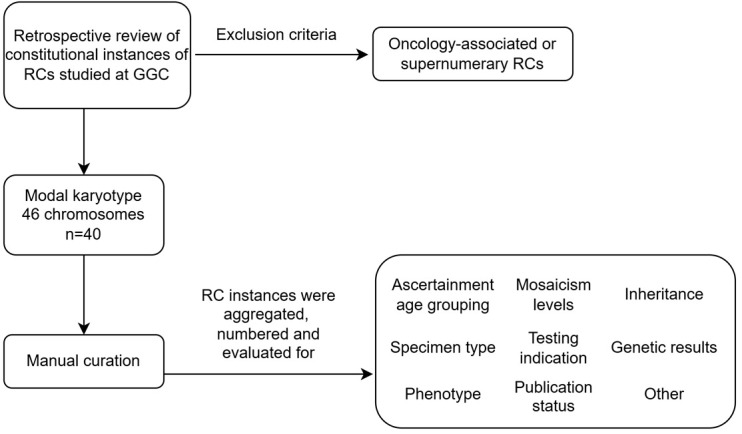
Overview of the RC curation process. Total RCs identified in a modal karyotype of 46 chromosomes are collected and curated at the single chromosome level. Where available, information collected during this process is depicted above.

**Figure 2 genes-16-00736-f002:**
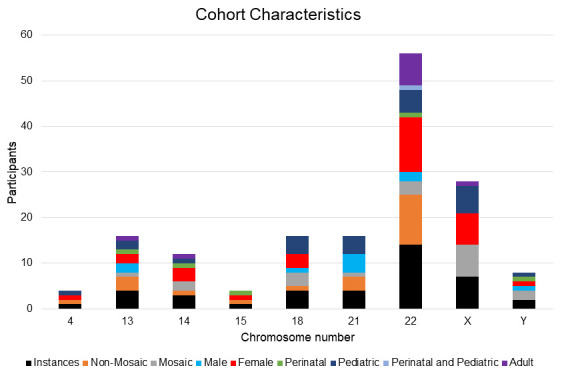
Total RCs are depicted at the single chromosome level, and both mosaic and non-mosaic forms are observed. Most were female and pediatric-aged at the time of ascertainment (<18 years old).

**Figure 3 genes-16-00736-f003:**
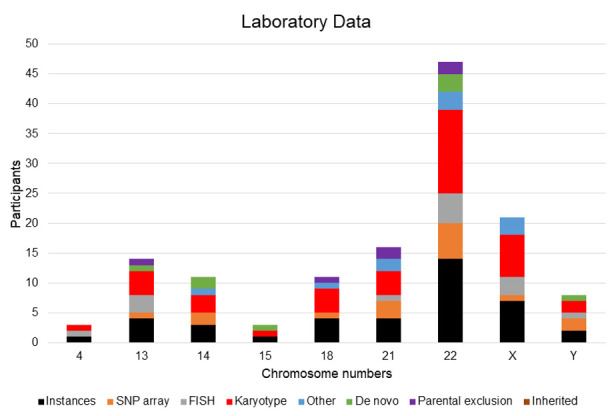
Laboratory data for the RCs are presented at the single-chromosome level. The majority were karyotyped, and many would benefit from further clarification of their breakpoints. A minority were proven de novo, and a subset had parental exclusion carried out, and none were inherited.

**Figure 4 genes-16-00736-f004:**
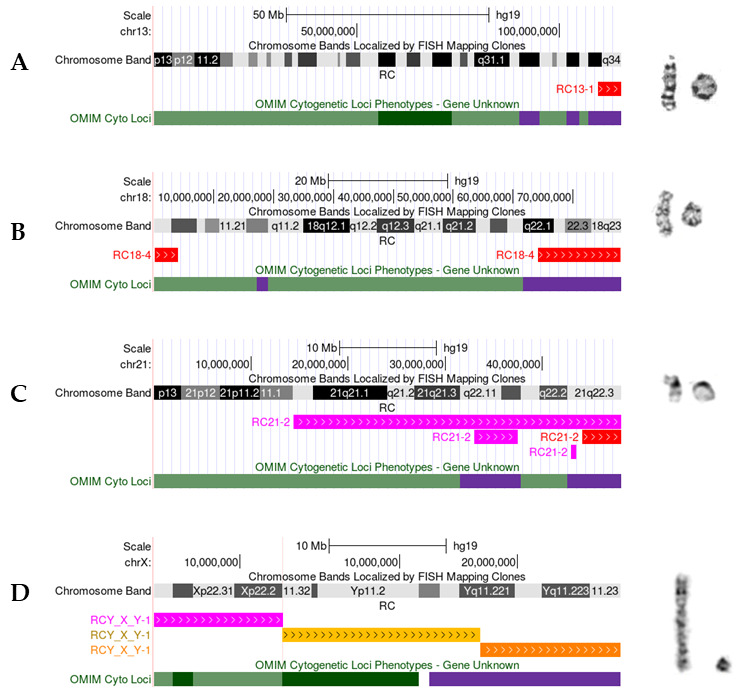
Selected RCs with CMA data (based on UCSC Genome Browser; hg19). Each panel represents individuals identified as bearing an RC displaying the copy number changes identified by CMA on the left and the partial G-banded karyotype with the normal homologue on the left and the RC on the far right. Copy number states are depicted: 0, (orange), 0~1 (amber), 1 (red), 1~2 (hot pink), 3 (blue). (**A**) simple terminal mechanism is exemplified by RC13-1 which displays a simple terminal loss of 13qter; (**B**) classic mechanism (p and q involvement) is typified by RC18-4 which demonstrates terminal loss of both arms; (**C**) complex intrachromosomal rearrangement resulting in RC21 is demonstrated by combinations of terminal and interstitial copy number changes in the long arm for RC21-2; and (**D**) complex interchromosomal rearrangement resulting in a derivative RCY shown here using the multi-region view which shows a mosaic gain of Xp material that has translocated to Y with a corresponding loss of the terminal Yp and Yq arms.

**Table 1 genes-16-00736-t001:** Details for autosomal RCs. A legend for abbreviations is available below the table.

RC	Type	Age	Gender	Sample	Features	Assay	Mosaic (K)	Inheritance	Status
RC4-1	CCR	P	F	PB	micrognathia, thin lips, slanted palpebral fissures down, helix abn, nasal tip abn, prematurity, congenital heart defect, advanced bone age, preauricular appendage, short philtrum (absent), microcephaly, midface hypoplasia, ID-severe	K, FISH	N	Unk	F;U
RC13-1	S	P	M	PB	DD, speech delay, microcephaly, café au lait spots, involuntary gait/movement, sloping forehead, dysmorphic facies, clinodactyly-fifth finger	K, FISH, CMA 13q33.3q34 loss (5.7 Mb)	N	Maternal exclusion	
RC13-2	S	A	F	PB	ID-mild, tonic-clonic (grand mal) seizure, depression	K, FISH	N	Unk	
RC13-3	S	P	M	PB	Hypotonia and microcephaly	K	93%	Unk	unstable RC
RC13-4	S	Fetus-stillborn	F	AF/Placental biopsy	Abn U/S (not NTD), cystic hygroma AF-AFP at risk AChE negative, triple screen negative, 2nd pregnancy, 17.4 weeks, affected by holoprosencephaly and MCA 23 weeks GA, TOF, hypoplastic external genitalia, cystic hygroma, thumb a/hypoplasia, renal a/hypogenesis, choanal atresia, lung a/hypoplasia, ectopic anus	K, FISH	6.67~13%	De novo	F;U
RC14-1	S	A	F	PB	ID-moderate and seizures, palpebral fissure abn, seizures, hyperpigmented macule	K, CMA 14q32.33 loss (4.7 Mb)	86%	Unk	
RC14-2	S	Fetus-POC	F	POC		K	63%	De novo	
RC14-3	S	P	F	PB	seizure disorder	K, CMA 14q32.33 loss (2.6 Mb)	N	De novo	
RC15-1	S	Fetus-POC	F	Fetal tissue		K	N	De novo	
RC18-1	S	P	F	PB	DD, affected by ring autosomes, ID-moderate, flat midface, slanted palpebral fissures-up, low nasal bridge depressed, hypoplastic alae nasi, flat midface, attached earlobes, heart murmur, short stature, telecanthus	K	N	Unk	
RC18-2	S	P	F	PB	ptosis of eyelid, eye, slanted palpebral fissures (up), protruding ears (lop ears), oral/mouth anomaly/saliva, hearing loss/impairment, DD	K, MLPA	52%	Unk	
RC18-3	S	P	M	PB	FTT, cleft palate	K	44%	Unk	
RC18-4	S	P	F	PB	DD, short stature, balance and coordination problems, large lobes, low-set ears, sloping forehead, upslanted palpebral fissures, brachydactyly, camptodactyly, long facies, midface hypoplasia, retrognathia, downturned corners of mouth	K, CMA 18p11.32p11.31 (3.8 Mb) and 18q22.1q23 (13.9 Mb) losses	85%	Maternal exclusion	
RC21-1	S	P	M	PB	dolichocephaly, ID-mild, attention problems, small earlobes, pes planus, midface hypoplasia, family history of ID, high arched palate, large nose, affected by ring autosome	K	N	Unk	
RC21-2	CCR	P	M	PB	holoprosencephaly, pulmonary hypertension	K, CMA 21q11.2q22.3 mosaic loss with mosaic gains of 21q22.11 (4.5 Mb) and 21q22.3 (589 Kb), and 21q22.3 loss (4 Mb)	70%	Maternal exclusion	
RC21-3	S	P	M	PB	Speech delay and hearing impairment, hyperactivity, DD, expressive speech delay/disorder, epicanthal folds, slanted palpebral fissures up, midface hypoplasia, thin nails	K, MLPA, FISH, qPCR, CMA 21q22.13q22.3 gain (10.2 Mb)	N	Maternal exclusion	
RC21-4	S	P	M	PB	Dysmorphic features, DD. Findings: coloboma, protruding ears (lop ears), cupped ears, microcephaly, ID-moderate, thin underweight, hyperreflexia, hypertonia	K, CMA 21q22.13 loss (9.6 Mb)	N	Unk	
RC22-1	S	A	F	PB	ID and seizure disorder	K, FISH	N	Unk	
RC22-2	S	A	F	PB	ID-profound, dysmorphic facies, obesity, microcephaly, Madelung deformity, sloping forehead, midface hypoplasia, slanted palpebral fissures up, sparse (absent) eyebrows, cataract	K	N	Unk	
RC22-3	S	A	F	PB	ID-severe, affected by neurofibromatosis (AD), flat midface, micrognathia, attached earlobes	K, FISH	N	Unk	
RC22-4	S	P	F	PB	Midface hypoplasia, fine and gross motor incoordination, thin finger nails, significant speech hard to understand, DiGeorge	K, FISH	61%	Paternal exclusion	
RC22-5	S	A	M	PB		K, CMA (negative)	8%	De novo	
RC22-6	S	P	F	PB	DD, syndactyly, hooded eyelids	K, CMA 22q13.33 loss (1.6 Mb)	N	De novo	
RC22-7	S	A	F	PB	Nose, ID, sunken/recessed eyes, small palpebral fissures, synophrys, macrotia, camptodactyly	K, FISH	N	Unk	
RC22-8	S	P	F	PB		K, CMA 22q13.31 (3.8 Mb) loss	N	De novo	
RC22-9	S	A	F	PB	hypopigmented macule, heart murmur, asymmetric facies, ID-moderate	K	N	Unk	
RC22-10	S	Fetus	F	CVS	AMA, AFP normal, 16.1 weeks, prior pregnancy normal male	K	N	Unk	
RC22-11	S	A	F	PB		K, CMA 22q13.2 loss (7.6 Mb)	N	Unk	
RC22-12	S	Fetus AND P	F	AF AND PB	quad screen negative, prenatal screen risk of Down syndrome, AF-AFP/MCA at birth	K, FISH	24% prenatal K and absent postnatal K	Unk	F;U
RC22-13	S	P	M	PB		K, CMA 22q13.2q13.31 gain (2.34 Mb) and 22q13.31qter loss (6.1 Mb)	N	Paternal exclusion	
RC22-14	S	P	F	PB	studied for 22q13 deletion	K, CMA 22q13.31 loss (4.4 Mb)	N	Unk	

Abbreviations: A; Adult, Abn; abnormality, AF; amniotic fluid, AFP; α fetoprotein, AMA; advanced maternal age, CCR; complex chromosomal rearrangement, DD; developmental delay, F; female, FU; Follow-up studies performed, FTT; failure to thrive, GA; gestational age, ID; intellectual disability, K; karyotype, Kb; kilobase, M; male, MB; megabases, MCA; multiple congenital anomalies, N; no, nb; newborn, NIPT; non-invasive prenatal (screening) testing, Nd; not determined, NTD; neural tube defect, P; Pediatric, PB; peripheral blood, POC; product of conception, S; Simple, SGA; small for gestational age, TOF; tetralogy of Fallot, Unk; unknown.

**Table 2 genes-16-00736-t002:** Details for gonosomal RCs. A legend for abbreviations is available below the table.

RC	Type	Age	Gender	Sample	Features	Assay	Mosaic (K)	Inheritance	Status
RCX-1	S	P	F	PB	possible TS, short stature, hyperconvex nails, sacral dimple, unfurled helices, epicanthal folds, upslanted palpebral fissures, cubitus valgus, metacarpal hypoplasia, acanthosis nigricans, growth hormone supplementation, which was discontinued. Mild vision issues, bilateral knee dislocations, Tanner stage IV pubic hair	K	4%	Unk	
RCX-2	S	A	F	PB	rule out mosaicism 45,X/46,XX or XY, diagnosed with mosaic TS at a young age, aberrant right subclavian artery CT scan discovered incidentally, hyperlipidemia, insomnia, premature menopause in early 20s due to TS	K	9.30%	Unk	
RCX-3	S	P	F	PB	ID, epicanthal folds, fifth finger a/hypoplasia, palmar line, short stature, DD, slanted palpebral fissure-up, ptosis of eyelid	K, FISH, DNA studies	8~11%	Unk	F;U; unstable RC
RCX-4	S	P	F	PB	TS, short stature, hypothyroidism, intellectual disability-unk, everted eyelid, palpebral fissure long, synophrys, arched eyebrows	K	33~40%	Unk	F;U
RCX-5	S	P	F	PB	short stature, FTT, undergrowth, SGA, TS	K, CMA (inconclusive)	28%	Unk	
RCX-6	S	P	F	PB	concern for 45,X mosaicism, abn sex chromosomes	K, CMA Xp11.22p11.21 mosaic gain (911 Kb) and Xp21q21.1 mosaic loss (25 Mb)	56%	Unk	
RCX-7	S	P	F	PB	rule out chromosomal abnormality	K, FISH	66%	Unk	
RCY-1	S	P	M	PB	bifid uvula, submucous cleft palate, ring chromosome Y syndrome	K, CMA Yp11.32 loss (1.7 Mb) and Yq12 loss (302 Kb), and Yq11.21q12 loss (15 Mb)	90%	De novo	
RCX;Y-1	CCR	Fetus	F	AF	AFP not elevated. NIPT high risk for monosomy X. GA 16 weeks	K, FISH, CMA Yp11.32q11.221 mosaic loss (17 Mb) and Yq11.221q11.23 loss (11.9 Mb) with Xp22.33p22.2 mosaic gain (11 Mb)	54%	Unk	

Abbreviations: A; Adult, Abn; abnormality, AF; amniotic fluid, AFP; α-fetoprotein, AMA; advanced maternal age, CCR; complex chromosomal rearrangement, DD; developmental delay, F; female, FU; Follow-up studies performed, FTT; failure to thrive, GA; gestational age, ID; intellectual disability, K; karyotype, KB; kilobase, M; male, MB; megabases, MCA; multiple congenital anomalies, N; no, nb; newborn, NIPT; non-invasive prenatal (screening) testing, Nd; not determined, NTD; neural tube defect, P; Pediatric, PB; peripheral blood, POC; product of conception, S; Simple, SGA; small for gestational age, TOF; tetralogy of Fallot, TS; Turner syndrome, Unk; unknown.

## Data Availability

De-identified data is available from the corresponding author upon reasonable request.

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
