# Peer review of "Characterization of Constitutional Ring Chromosomes over 37 Years of Experience at a Single-Site Institution"

_genes, 2025, doi:10.3390/genes16070736_

Round 1

Reviewer 1 Report

Comments and Suggestions for Authors

The manuscript is generally well written and presents valuable data that will be of interest to geneticists and other clinicians who encounter patients with ring chromosomes. However, the information needs to be presented more clearly for easier reading and reference (please see my comments below). 

Major concern:

The reported material includes a large amount of information that is difficult to follow in the text for each case. Creating a more comprehensive table to replace Table 1 may help increase the clarity of the report. Such a table should include the type of RC (with the size of the deleted material), if mosaic, the degree of mosaicism should be reported, the age of diagnosis, gender, status of the fetus if the diagnosis was made prenatally: live-born vs. lost pregnancy, inheritance, and summary of the clinical features.

Alternatively, if there are too many cases to fit in one table, separate tables should be created for each RC set.

Minor concerns

Figure 1

For all graphs, suggest labelling the X axis as “chromosome number” for clarity

1B: age of ascertainment is not indicated on the graphs but is mentioned in the Figure legend. Please indicate it on the graph or remove it from the text in the legend.

1D parental exclusion line is not clear

Figure 2-in the legend, please specify the color coding, for example, red=deletion, etc.

For easier reference, please replace in the text sentences like “ only the child” or “in the adult” with the number of the individual, like RC14-1, RC14-1

Author Response

Comments and Suggestions for Authors

REVIEWER 1

Major concern:

The reported material includes a large amount of information that is difficult to follow in the text for each case. Creating a more comprehensive table to replace Table 1 may help increase the clarity of the report. Such a table should include the type of RC (with the size of the deleted material), if mosaic, the degree of mosaicism should be reported, the age of diagnosis, gender, status of the fetus if the diagnosis was made prenatally: live-born vs. lost pregnancy, inheritance, and summary of the clinical features.

Alternatively, if there are too many cases to fit in one table, separate tables should be created for each RC set.

We thank the reviewer for this feedback. We have created two comprehensive tables (one for autosomes and one for sex chromosomes) to increase the clarity of the report. The table includes details on the type of RC, size of the deleted material, mosaicism, degree of mosaicism, age grouping at diagnosis, gender, status, inheritance, and a summary of clinical features.

Minor concerns

Figure 1

For all graphs, suggest labelling the X axis as “chromosome number” for clarity

We have made an edit to the X-axis.

1B: age of ascertainment is not indicated on the graphs but is mentioned in the Figure legend. Please indicate it on the graph or remove it from the text in the legend.

We have removed this phrasing from the text in the legend and replaced it with age grouping references instead.

1D parental exclusion line is not clear

 We have changed from lines to clustered bar graphs instead.

Figure 2-in the legend, please specify the color coding, for example, red=deletion, etc.

 We have specified color coding and corresponding copy number in the legend.

For easier reference, please replace in the text sentences like “ only the child” or “in the adult” with the number of the individual, like RC14-1, RC14-1

 We have swapped references using individual numbers instead of text sentences.

Reviewer 2 Report

Comments and Suggestions for Authors

This retrospective study represents a comprehensive, longitudinal look at constitutional ring chromosomes and provides a significant resource to the genetics community.  It reinforces there are previously known hotspots for ring chromosome formation and highlights the strengths of combining karyotype and microarray analyses in characterization of ring chromosomes.  The detailed case study highlighting complex rearrangements further illustrates the importance of integrating multiple methodologies to accurately characterize the chromosomes and for follow-up familial testing. 

Some minor suggestions to improve the manuscript would be to reformat Table 1 (phenotypic heatmap) so that the number of cases are centered in the column.  As it is currently presented, the format makes it difficult to associate the number of cases with specific phenotypes. 

Also could the authors comment on whether another tissue type was tested for any of the 40 cases?  If so, did the structure of the ring vary?  How did the percentage of cells with the ring chromosome vary?  

Overall an outstanding study looking at the frequency and impact of ring chromosomes.  

Author Response

Comments and Suggestions for Authors

Reviewer 2

This retrospective study represents a comprehensive, longitudinal look at constitutional ring chromosomes and provides a significant resource to the genetics community.  It reinforces there are previously known hotspots for ring chromosome formation and highlights the strengths of combining karyotype and microarray analyses in characterization of ring chromosomes.  The detailed case study highlighting complex rearrangements further illustrates the importance of integrating multiple methodologies to accurately characterize the chromosomes and for follow-up familial testing. 

Some minor suggestions to improve the manuscript would be to reformat Table 1 (phenotypic heatmap) so that the number of cases are centered in the column.  As it is currently presented, the format makes it difficult to associate the number of cases with specific phenotypes. 

Based on feedback from multiple reviewers, we have replaced the heatmap with case-level tables that include details to facilitate the association between case descriptions and conclusions.

Also could the authors comment on whether another tissue type was tested for any of the 40 cases?  If so, did the structure of the ring vary?  How did the percentage of cells with the ring chromosome vary?  

We have added sample type information to the tables and commented on instability where known, as well as levels of mosaicism.

Overall an outstanding study looking at the frequency and impact of ring chromosomes.  

Reviewer 3 Report

Comments and Suggestions for Authors

Thank you for the opportunity to review the manuscript “Characterization of Constitutional Ring Chromosomes- Over 37 Years of Experience at a Single-Site Institution” by Jaclyn B. Murry and Barbara G. DuPont. It refers to an interesting issue of genetic abnormalities in a form of constitutional ring chromosomes and their clinical phenotype. However, there are some issues that need to be improved.

  1. The Introduction is too short, there is lack of clearly formulated aim of the study.
  2. Which years does the analysis include?
  3. 2. RC Summary- please do not use abbreviations in the subtitles.
  4. Results are difficult to follow- text needs to be shortened, some results would be better to present in a table/graph/figure. Some supplementary figures could be implemented into the main manuscript.
  5. Limitations of the study should be included in the Discussion not Conclusions.
  6. Conclusions should be more specific.

In my opinion the weakest point of the manuscript is lack of presenting the actual aim of the study and therefore lack of precise conclusions. Description of different types of constitutional ring chromosomes seems not to be enough.

Comments on the Quality of English Language

The English could be improved to more clearly express the research.

Author Response

Comments and Suggestions for Authors

Reviewer 3

Thank you for the opportunity to review the manuscript “Characterization of Constitutional Ring Chromosomes- Over 37 Years of Experience at a Single-Site Institution” by Jaclyn B. Murry and Barbara G. DuPont. It refers to an interesting issue of genetic abnormalities in a form of constitutional ring chromosomes and their clinical phenotype. However, there are some issues that need to be improved.

  1. The Introduction is too short, there is lack of clearly formulated aim of the study.

We thank the reviewer for this feedback and have strengthened the introduction and added detail to our study aim.

  1. Which years does the analysis include?

Due to differences in laboratory information systems, we can deduce that we have included RC cases from the late 1980s to the present.

  1. 2. RC Summary- please do not use abbreviations in the subtitles.

We thank you for this comment and have changed the subtitles to non-abbreviations.

  1. Results are difficult to follow- text needs to be shortened, some results would be better to present in a table/graph/figure. Some supplementary figures could be implemented into the main manuscript.

We thank the reviewer for their feedback and have removed some text and converted the phenotype heatmap into two tables to aid in the understanding of our manuscript. If there are specific supplemental figures that would benefit from their inclusion in the main manuscript, we would be glad to make these modifications.

  1. Limitations of the study should be included in the Discussion not Conclusions.

We thank you for pointing this out and have moved the limitations to discussion.

  1. Conclusions should be more specific.

We have enhanced the conclusions and hope this addresses the reviewer’s concern.

In my opinion the weakest point of the manuscript is lack of presenting the actual aim of the study and therefore lack of precise conclusions. Description of different types of constitutional ring chromosomes seems not to be enough.

We have attempted to clarify the aim of the study and precise conclusions, and hope we have addressed the reviewer’s concerns.

Comments on the Quality of English Language

The English could be improved to more clearly express the research.

We have attempted to cut some areas of wordiness and hope this aids clarity in the manuscript.

Reviewer 4 Report

Comments and Suggestions for Authors

Although the article is well-planned and grammatically sound, I do have the following suggestions to improve the quality of the manuscript. 

A. The introduction needs to be improved by adding a few paragraphs and references from 2020-2023. 

B. Introduction should include the primary and secondary aims of the study. 

C. Flow diagram for the various steps of the study should be included for clarity. 

D. Drawbacks or limitations should be well written in a paragraph. 

E. It will be great to include a high-resolution diagram for Fig. 2.

Author Response

Comments and Suggestions for Authors

Reviewer 4

Although the article is well-planned and grammatically sound, I do have the following suggestions to improve the quality of the manuscript. 

  1. The introduction needs to be improved by adding a few paragraphs and references from 2020-2023. 

We thank the reviewer for this feedback and have added content to the introduction as well as included more recent references.

  1. Introduction should include the primary and secondary aims of the study. 

We appreciate this comment and have added the primary and secondary aims of this study to the introduction.

  1. Flow diagram for the various steps of the study should be included for clarity. 

We appreciate this comment and have added a flow chart to the main manuscript to explain the methodology used in this study.

  1. Drawbacks or limitations should be well written in a paragraph. 

We appreciate this concern and have sought to improve the limitation in the discussion and hope this addresses the reviewer’s concern.

  1. It will be great to include a high-resolution diagram for Fig. 2.

We have recreated this figure by simplifying the content, and we hope this improves the resolution for this reviewer.

Round 2

Reviewer 3 Report

Comments and Suggestions for Authors

Thank you for the improvement od the manuscript. I would recommend improving the graphics to make them more readable.

Comments on the Quality of English Language

The English could be improved to more clearly express the research.

Author Response

Thank you for the suggestions. We have cut portions of the text and have reimported graphics as tiffs to make them more readable.  We hope this addresses this reviewer's concern.